# New Concept for the Facile Fabrication of Core–Shell CuO@CuFe_2_O_4_ Photocathodes for PEC Application

**DOI:** 10.3390/ma15031029

**Published:** 2022-01-28

**Authors:** Linh Trinh, Krzysztof Bienkowski, Piotr Wróbel, Marcin Pisarek, Aleksandra Parzuch, Nabila Nawaz, Renata Solarska

**Affiliations:** 1Laboratory of Molecular Research for Solar Energy Innovations, Centre of New Technologies, University of Warsaw, 02-097 Warsaw, Poland; k.bienkowski@cent.uw.edu.pl (K.B.); a.parzuch@student.uw.edu.pl (A.P.); n.nawaz@cent.uw.edu.pl (N.N.); 2Faculty of Physics, University of Warsaw, 02-093 Warsaw, Poland; piotr.wrobel@fuw.edu.pl; 3Institute of Physical Chemistry, Polish Academy of Sciences, 01-224 Warsaw, Poland; mpisarek@ichf.edu.pl

**Keywords:** metal oxides, spinel metal oxides, Prussian blue analogues, photoelectrochemistry, electrodeposition

## Abstract

The CuO@CuFe_2_O_4_ core–shell structure represents a new family of photocatalysts that can be used as photoelectrodes that are able to produce hydrogen under a broad spectrum of visible light. Herein, we report a novel approach for the production of this active film by the thermal conversion of CuFe Prussian Blue Analogues. The outstanding photoelectrochemical properties of the photocathodes of CuO@CuFe_2_O_4_ were studied with the use of combinatory photo-electrochemical instrumental techniques which proved that the electrodes were stable over the whole water photolysis run under relatively positive potentials. Their outstanding performance was explained by the coupling of two charge transfer mechanisms occurring in core–shell architectures.

## 1. Introduction

Due to an increasing demand for and the limitation of fossil fuels, hydrogen fuel produced by photoelectrochemical (PEC) water splitting has been considered as a promising alternative since it is known for being clean, abundant and sustainable [1,2,3]. Copper ferrite (CuFe_2_O_4_) has been studied for its catalytic H_2_ production owing to its narrow bandgap of 1.3–2 eV with a conduction band edge above 0 V vs. RHE [4]. Despite this, CuFe_2_O_4_ has mostly been studied as photoanodes [5,6,7,8,9]. There are some studies of CuFe_2_O_4_ as photocathodes for H_2_ production and CO_2_ reduction, however, most studies on these materials still have a number of challenges as CuFe_2_O_4_ suffers from severe photocorrosion and/or low photocurrent densities [10,11,12,13,14]. It has been reported that the combination of p-type Cu and α-Fe_2_O_3_ would be favorable for water splitting due to the shift in the conduction band edge potential, leading to a greater absorption of photons in the visible spectrum [15,16]. Recently, Park et al. [17] reported the rapid flame-annealed CuFe_2_O_4_ with an exceptional photocurrent density of −1.82 mA/cm^2^ at 0.4 V RHE despite its short-term stability and heavy photocorrosion. Maitra et al. [18] proposed a wet chemistry synthetic route for a highly porous CuFe_2_O_4_ nanoflake where the degrees of spinel inversion (δ) of the materials were taken into account and CuFe_2_O_4_ prepared at medium temperature (230 °C) showed the highest photocurrent density at 0 V vs. RHE with ***J*** = −0.99 mA/cm^2^; however, its stability was not reported.

In the frame of this work, we propose a new spinel CuFe_2_O_4_ prepared by thermal conversion from CuFe Prussian Blue Analogues (PBAs) (formula: Cu^II^[Fe^III^(CN)_6_]·*x*H_2_O). PBAs are the derivative forms of Prussian Blue (PB) with the general formula K_4_[Fe^III^Fe^II^(CN)_6_] prepared by replacing one of the Fe ions by another transition metal [19]. These materials have a historical figure owing to their unique electronic and optical properties. The thermal conversion of PBAs into spinel metal oxides was first proposed by Zakaria et al. [20] on FeCo PBAs, CoCo PBAs, and PB by annealing them at high temperature to eliminate the C≡N bonds between the two metal centers to form spinel CoFe_2_O_4_, Co_3_O_4_, and Fe_2_O_3_, respectively. This conversion method has the advantage of retaining the original structure of PBAs as they possess an abundance of metal ions and high porosity. The incorporation of porous structure and noble metals into semiconductors offer a promising strategy for fabricating a structure with greater light activity due to a larger surface area which serves as a support to bind particles [21,22].

In this study, CuFe PBAs thin film was deposited on FTO glass using electrosynthesis and then annealed at 550 ${}^{\circ }$C to form a CuFe_2_O_4_ photocathode. The photocathode had a sufficiently high photocurrent density of −0.3 mA/cm${}^{2}$ with a small photocorrosion observed. Moreover, the materials were shown to have great stability over time. The materials were investigated by various techniques.

## 2. Materials and Methods

### 2.1. Preparation of CuO@CuFe_2_O_4_

The layer of CuO@CuFe_2_O_4_ on FTO glass (dimension 15 mm × 35 mm) was prepared by electrodepostion performed on Biologic SP-300 potentiostat using a conventional three-electrode system with an Ag/AgCl reference electrode, Pt wire as the counter electrode and clean FTO glass as the working electrode. First, the working electrode was applied in a constant potential of −0.9 V (vs. Ag/AgCl) for 4 min in an aqueous solution of 80 mL containing 10 mM CuSO_4_ (CuSO_4_· 5H_2_O Sigma-Aldrich 99.99% CAS: 7758-99-8), 100 mM K_2_SO_4_ (K_2_SO_4_ Sigma-Aldrich ≥ 99.0% CAS: 77778-80-5), and 1 mM H_2_SO_4_ for the deposition of Cu. After that, the Cu was cleaned with water and was applied at a constant potential of 0.5 V (vs. Ag/AgCl) for 30 min in an aqueous solution of 80 mL containing 10 mM K_4_[Fe(CN)_6_] · 3H_2_O (Sigma-Aldrich 98.5% CAS: 13746-66-2), 50 mM KCl (Sigma-Aldrich 99.0% CAS: 7447-40-7), 50 mM K_2_SO_4_, and 1 mM H_2_SO_4_ for the formation of Cu[Fe(CN)_6_]. After a few minutes, the layer changed color from red to yellow, which is typical of Cu[Fe^III^(CN)_6_]. After the electrosynthesis, the FTO glass was cleaned thoroughly with water and acetone and put into a furnace and annealed for one hour at 400 ${}^{\circ }$C to form CuO@CuFe_2_O_4_. Then, the electrode was put into an oven and annealed again for 10 h at 550 ${}^{\circ }$C for recrystallization.

### 2.2. Characterization Methods

The morphology of CuO@CuFe_2_O_4_ arrays was examined by SEM using a Carl Zeiss Sigma HV workstation (GmbH, Oberkochen, Germany). The microscope was equipped with a Gemini electron column with an energy-selective backscattered detector and energy-dispersive X-ray spectrometer with Bruker Quantax XFlash 6|10 detector (GmbH, Karlsruhe, Germany). UV–Vis spectroscopy was conducted using a Jasco V-650 spectrophotometer equipped with a 60 mm integrating sphere (Jasco, Easton, MD, USA). Size distribution was performed using ImageJ software (https://imagej.nih.gov/ij/, accessed on 30 December 2021). Powder X-ray diffraction was performed on samples on FTO glass and data were collected on an X’Pert PRO MPD powder diffractomer (manufactured by Panalytical B.V. Netherlands) using Co–Kα ( with Fe filter) radiation equipped with a fast detector. Then, 2θ was converted to that of Cu-Kα. The crystal structure was refined based on data obtained from the XRD pattern using the VESTA 3 software, National Museum of Nature and Science, 4-1-1, Amakubo, Tsukuba-shi, Ibaraki 305-0005, Japan [23]. XPS measurements were performed using a Microlab 350 (Thermo Electron, East Grinstead, UK) spectrometer, which was equipped with a dual Al/Mg anode. The X-ray radiation source (Al-Kα ) at 1486.6 eV was used for investigations of the following parameters: power 300 W, voltage 15 kV, emission current 20 mA. All the XPS spectra for individual elements were recorded at a pass energy 40 eV, energy step size 0.1 eV. Avantage software (Version 5.9911, Thermo Fisher Scientific, Waltham, MA, USA) for data processing was used to perform the deconvolution procedure by using an asymmetric Gaussian/Lorentzian mixed function at a constant G/L ratio equal 0.35 (±0.05). The measured binding energies were corrected in reference to the energy of C 1s peak at 285.0 eV.

### 2.3. PEC Measurement

A conventional three-electrode system consisting of CuO@CuFe_2_O_4_ deposited on FTO glass as the working electrode, a platinum wire as the counter electrode, and an Ag/AgCl reference electrode in saturated KCl solution were implemented in 0.1 M NaOH electrolyte solution (pH 13). The electrolyte was purged with N_2_ for 35 min before every measurement to remove O_2_. Electrochemical potentials were converted to the RHE scale ($E_{\left(RHE\right)}=E_{\left(\mathrm{Ag}/\mathrm{AgCl}\right)}+E_{\left(\mathrm{Ag}/\mathrm{AgCl}\right)}^{0}+0.059\times pH$ where $E_{\left(\mathrm{Ag}/\mathrm{AgCl}\right)}^{0}=0.175$ V and pH = 13). The working electrodes were polarized at 10 mV/s by a CHI660D potentiostat. Simulated AM 1.5 G (100 m·W·cm${}^{2}$) illumination was obtained with an Oriel 150 W solar simulator (LoT Quantum Design, Darmstadt, Germany). The IPCE vs. excitation wavelength graphs were obtained using light from a 500 W Xenon lamp and a Multispec 257 monochromator (Oriel) with a typical bandwidth of 4 nm. The absolute light intensity passing through the monochromator was measured with an OL 730-5C UV-enhanced silicon detector (Gooch& Housego, Darmstadt, Germany). The current versus potential (**J**-**E**) plots of CuO@CuFe_2_O_4_ photocathodes were measured in a Teflon cell equipped with a quartz window. The exposed CuO@CuFe_2_O_4_ electrode surface area was 0.28 cm${}^{2}$.

## 3. Results and Discussion

### 3.1. Characterization

The morphology of CuFe PBAs and CuFe_2_O_4_ after electrodeposition and heat treatment were characterized by scanning electron microscope (SEM), as shown in Figure 1. Before annealing, the sample consisted of cubic nanoparticles (NPs) typical of PBAs [19]. The layer of $Cu[Fe{\left(CN\right)}_{6}]$ was observed to be dense and uniform thickness-wise due to the nature of the synthetic method, but not homogeneous. After annealing, the grains retained their density but changed in shape, from cubic NPs to amorphous shape with an average size of approximately 28.84 ± 4.03 nm (Appendix A), confirming the conversion of CuFe PBAs to CuFe_2_O_4_. The particles appeared to be more homogeneous size-wise than its starting materials due to the breakdown of larger Cu[Fe(CN)_6_] NPs caused by the elimination of the C≡N bonds. The thickness of the films remained mostly the same after thermal conversion. The energy-dispersive spectroscopy (EDS) analysis showed the ratio of the components before thermal treatment (CuFe_2_O_4_ (Appendix A) are Cu: 0.4; Fe: 0.24; C: 1.44; and N: 2.1, indicating the formation of the CuFe PBA and the excess of Cu was the unreacted residual. After annealing, the composition of the film was Cu: 0.41; Fe: 0.28; O: 1.2 confirming the composition of spinel oxides CuO@CuFe_2_O_4_ (Appendix A). The excessive amount of O in the structure was mostly due to the initial porosity of Cu[Fe(CN)_6_].

EDS mapping shown in Figure 2 revealed that Fe was positioned on the outer shell of the NPs while Cu and O could be observed all over the whole particles. This indicated that, during electrosynthesis, the outer shell of the Cu grain reacted with [Fe^III^(CN)_6_] to form Cu[Fe(CN)_6_] while the inner core remained Cu. In a typical PBA structure, defection sites are present where the vacancies of M-CN are completed by water molecules. Upon thermal treatment, these sites were most likely where the breaking down of the precursors Cu@Cu[Fe(CN)_6_] happened during the formation of the core–shell structures. The outer shell formed CuFe_2_O_4_ and the core formed CuO.

The formation of CuO@CuFe_2_O_4_ could be confirmed by X-ray diffraction (XRD) pattern as presented in Figure 3. The major diffraction peaks can be indexed as (220), (103), (311), (312), (511), and (440) of the tetragonal CuFe_2_O_4_ (JCPDS No. 34-0425) and the peaks of CuO (JCPDS No. 80-1917) at (111/200) and (202). All the peaks were well defined, indicating the good crystallinity of the film. The peaks of Fe_2_O_3_ (JCPDS No. 33-0664) at (012), (104), and (113) were also observed on the XRD pattern.

Grain sizes obtained by Scherrer equation from the XRD pattern:(1)$$D=\frac{k\times \lambda }{\beta \times \cos\theta }$$
where *k* = 0.94, λ = 1.54 Å, and β is the full-width half-maximum in radian, calculated to be 28.09 nm for the CuFe_2_O_4_ domain which was in great agreement with the grain sizes observed on SEM indicating its single crystallinity. The CuO domain was calculated to be 26.6 nm, which was slightly smaller than that of CuFe_2_O_4_. This again confirmed the core–shell structure of CuO@CuFe_2_O_4_.

The lattice constant of the tetragonal CuFe_2_O_4_ shell were calculated using the following relation [24]:(2)$$\frac{1}{d^{2}}=\frac{\left(h^{2}+k^{2}\right)}{a^{2}}+\frac{l^{2}}{c^{2}}$$
where *a* and *c* are lattice parameters, (*hkl*) is the miller indices, and *d* is the interplanar distance. The lattice parameter *a* value for the tetragonal CuFe_2_O_4_ shell was calculated to be 8.41 Å which is much greater than the reported values of ∼5.9 Å for tetragonal CuFe_2_O_4_ [24,25]. In addition, the *c* parameter of CuFe_2_O_4_ was determined to be 8.0 Å, slightly smaller than that of previously reported values of 8.4 Å [24,25]. This was due to the fact that the CuFe_2_O_4_ shell conserved most of the structure of PBAs, which had a face-centered cubic with a unit parameter at approximately 10 Å, making them excellent precursors for nanoporous metal oxides. Upon thermal conversion, the C≡N bonds were eliminated and substituted by O, leading to the shortening of the bond length between two metal centers.

Figure 4 shows the high-resolution core-level XPS spectra of the CuO@CuFe_2_O_4_ sample. The shape of the Cu2p spectrum (left) suggested that the copper was in an oxidized state. Apart from the Cu2p3/2 and Cu2p1/2 peaks, the satellite lines characteristic of ${\mathrm{Cu}}^{2+}$ in CuO are clearly visible [26,27]. The deconvolution procedure used for the copper peak confirms the presence of CuO (Cu2p3/2—932.7 eV and Cu2p1/2—952.5 eV) as well as the CuFe_2_O_4_ compound (Cu2p3/2—934.4 eV and Cu2p1/2—954.6 eV) [28,29]. The spectral line at 934.4 eV can also be attributed to copper hydroxide or copper sulfate [28,29], but the presence of these compounds is unlikely. This is due to the fact that the produced material was annealed twice after being prepared at 400 and 550 °C. The high resolution Fe2p spectrum (right) clearly indicates the presence of Fe oxide bonds in the investigated material, which can be assigned to the Fe${}^{3+}$ (710.6 eV, 724.1 eV) [30,31] and Fe${}^{2+}$ (708.9 eV and 722.6) [30,31] peaks. This was also confirmed by the shape of the iron Fe2p spectral line, where characteristic satellite lines from the detected oxides are visible [32]. For the recorded iron peak (Fe2p), the satellite lines are unusual. In particular, the lines that are located between the Fe2p3/2 and Fe2p1/2 peaks are too intense. This is due to the presence of a tin signal (Sn3p3/2) [31] at this point which comes from the sample substrate (FTO). Further considering the spectrum of Fe2p, the peaks at 712.5 and 726.2 eV might suggest the presence of copper ferrite [28,29]. Therefore, the obtained XPS results are consistent with the XRD measurements, which confirm the core–shell structure of the material.

### 3.2. Optical Properties

The UV–Vis spectra and Tauc plot of the direct bandgap of CuO@CuFe_2_O_4_ are presented in Figure 5. The bandgap energy of CuO@CuFe_2_O_4_ was found to be 1.5 eV larger than the value reported by Kezzim et al. [5] which was due to the presence of CuO in the core. The larger bandgap could also be due to the smaller crystal sizes of the NPs [33]. On the other hand, the UV–Vis spectra was shown to have an absorption band in the visible light range.

### 3.3. Photoelectrochemical Properties

Figure 6A shows the linear sweeping voltammetry (LSV) of the CuO@CuFe_2_O_4_ electrode in N_2_-saturated NaOH 0.1 M solution under chop light irradiation. The photoelectrode was shown to have a maximum photocurrent density of approximately −1 mA/cm${}^{2}$ at 0.3 V vs. RHE with a clear cathodic characteristic of p-type materials. Chronoamperometry measurements Figure 6B (right) at 0.5 V vs. RHE showed that the electrode had a photocurrent density of approximately −0.35 mA/cm${}^{2}$ with no corrosion observed, which is quite unusual for Cu-based materials [34]. Extended chronoamperometry measurements during one hour (Appendix A) showed that the electrode retained approximately 30% of its current density with almost no photocorrosion.

Mott–Schottky measurement of CuO@CuFe_2_O_4_ is presented in Figure 6C. The flatband potential ($E_{fb}$) of the materials was evaluated using Mott–Schottky analysis. A negative slope was observed for CuO@CuFe_2_O_4_ demonstrating p-type semiconductor behavior of the photoelectrode. For a p-type semiconductor, $E_{fb}$ is generally located near the valence band [35] and it can be estimated from the intersection of the plot of 1/*C*^2^ vs. E by the following equation:(3)$$\frac{1}{C^{2}}=\frac{2}{e\epsilon \epsilon_{0}N}\left(E-E_{fb}-\frac{kT}{e}\right)$$
where *C* is the capacitance, *e* is the electron charge, ϵ is the dielectric constant, $\epsilon_{0}$ is permittivity of vacuum, *N* is acceptor density, *E* is the electrode potential, *k* is the Boltzmann constant, and *T* is the temperature. $E_{fb}$ was estimated to be 1.27 V vs. RHE. Since the bandgap of CuO@CuFe_2_O_4_ was determined to be 1.5 eV, the band position was indeed in the range for $H_{2}$ production.

The impedance Nyquist plot in Appendix A clearly shows only one semicircle characteristic that proves that the charge transfer process occurs between the solid phase and the electrolyte. Therefore, the water reduction process takes place on the CuFe_2_O_4_ surface. The charge transfer resistance between phases of CuO and CuFe_2_O_4_ is negligible, which proves that the obtained core–shell structure holds a very good adhesion. If the water photoreduction process took place over CuO phase, a second semicircle of characteristics would be visible in the diagram. Thus, CuFe_2_O_4_ is a material that allows the transport of the generated charge without any significant losses because of charge recombination and charge accumulation at the surface, which in turn decreases the extent of CuO photocorrosion itself.

Figure 6D shows the *IPCE* spectrum of CuO@CuFe_2_O_4_. The %IPCE values were calculated using the following equation:(4)$$IPCE\left(\%\right)=1239.7\times \frac{J}{\lambda \times P}\times 100\%$$
where *J* is the photocurrent density, *P* is the intensity of the monochromatic light recorded with a power meter equipped with a thermopile detector and a calibrated silicon photodiode, and λ is the wavelength of the incident light. The %*IPCE* values reached their maximum value at approximately 420 nm and 490 nm with the value of 3.8%. The results showed that the %*IPCE* spectrum followed the trend of the UV–Vis absorption results.

## 4. Conclusions

In this study, core–shell CuO@CuFe_2_O_4_ was prepared using the facile method by thermal conversion from Cu@Cu[Fe(CN)_6_] precursors. After annealing, the photoelectrode exhibited p-type semiconductor characteristics. The CuO exhibits a p-semiconductor nature, which is strongly dependent on the delocalized hole states occurring in function of the concentration of the Cu vacancies [36]. The conductivity of such a system in general is poor due to relatively low electron concentration in the conduction band combined with slow carrier mobility. However, the charge transfer mechanism applied to CuFe_2_O_4_ was assigned to the typical small polaron hopping of SC conduction band “d” [37]. The photocurrent density under chop light irradiation at 0.5 V vs. RHE was −0.35 mA/cm^2^, which retained 50% of that after one hour, showing great stability. The electrode showed only a small photocorrosion indicating a decrease in the electrons–holes recombination usually observed for this type of materials. A closer look at the crystal structure (Appendix A) showed an increase in cell volume due to the porosity of starting materials. Upon the elimination of the C≡N bonds and the substitution of O, the bond lengths between Cu and Fe ions were slightly shorter due to the formation of Cu-O and Fe-O bonds.

## Figures and Tables

**Figure 1 materials-15-01029-f001:**
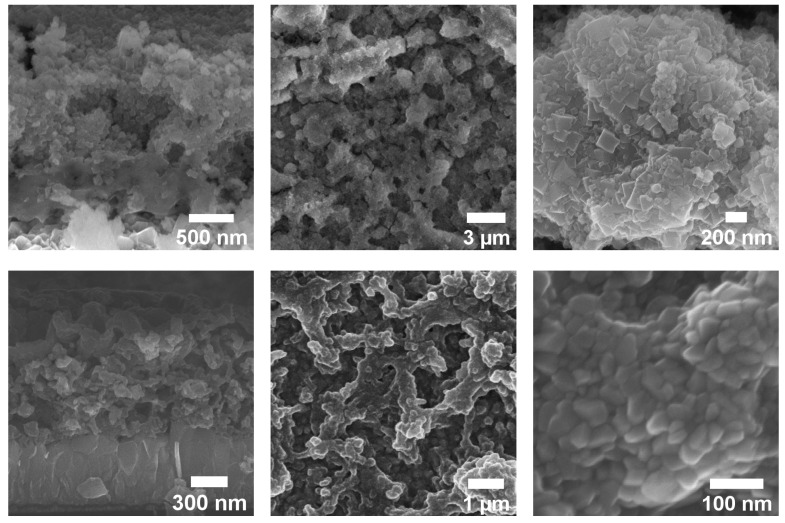
SEM imaging of Cu@CuFe PBAs (top images) and CuO@CuFe_2_O_4_ (bottom images).

**Figure 2 materials-15-01029-f002:**
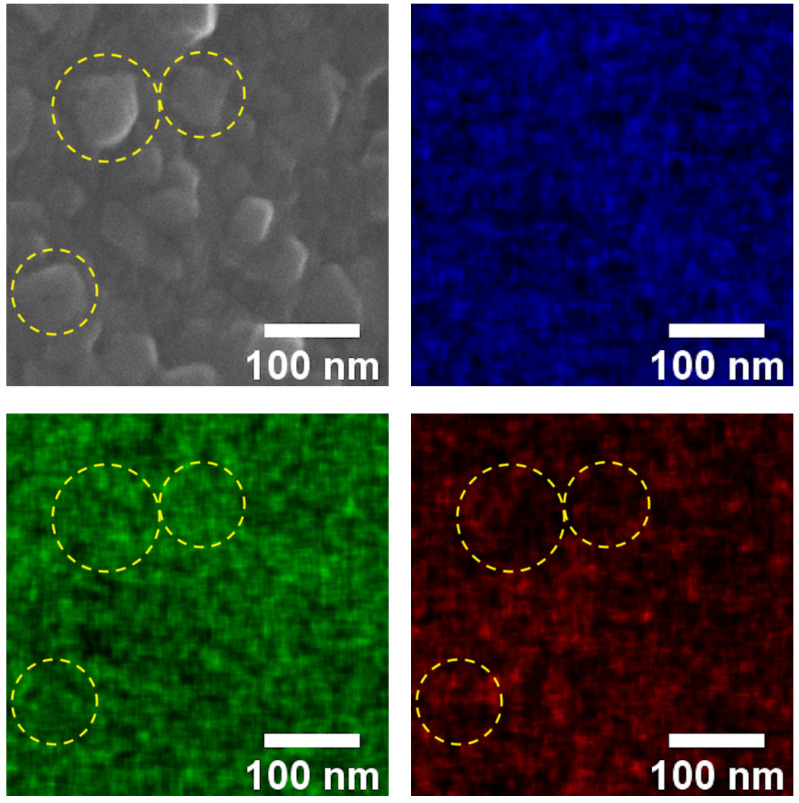
EDS mapping of Cu, Fe, and O on the CuO@CuFe_2_O_4_ layer. (mark circles indicated the mapping at the same positions).

**Figure 3 materials-15-01029-f003:**
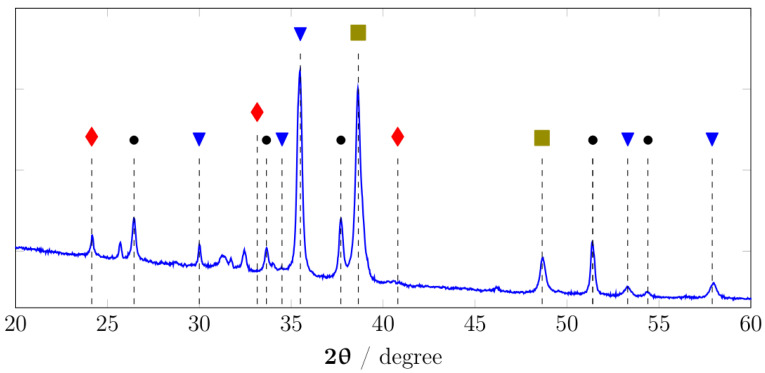
XRD patterns of CuO@CuFe_2_O_4_ (• = FTO, ▼ = CuFe_2_O_4_, ⧫ = Fe_2_O_3_, and ■ = CuO).

**Figure 4 materials-15-01029-f004:**
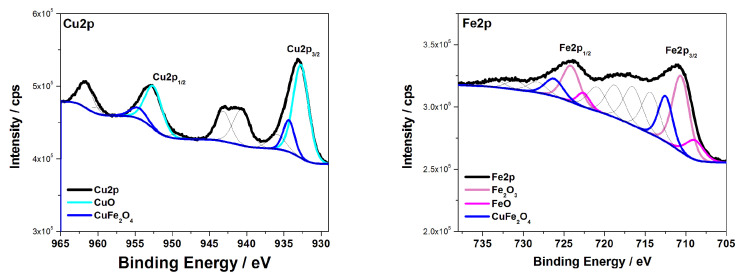
XPS high-resolution spectra of Cu2p (**left**) and Fe2p (**right**) for the CuO@CuFe_2_O_4_ sample after the deconvolution procedure.

**Figure 5 materials-15-01029-f005:**
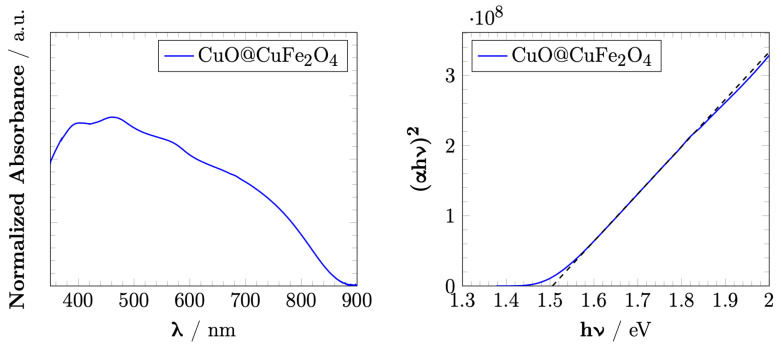
Normalized UV–Vis absorbance band of CuFe_2_O_4_ (**left**) and the Tauc plot of the direct optical band gap of CuFe_2_O_4_ (**right**).

**Figure 6 materials-15-01029-f006:**
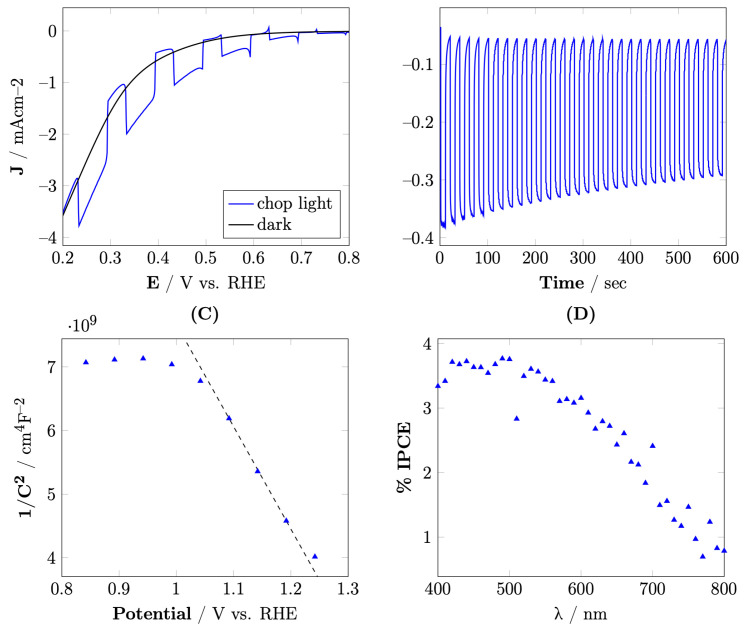
(**A**) Cathodic LSV scan of CuO@CuFe_2_O_4_ under chop light irradiation in 0.1 M NaOH (pH 13) with $N_{2}$ purged; (**B**) i-T curve of CuO@CuFe_2_O_4_ under chop light irradiation at 0.5 V vs. RHE in 0.1 M NaOH (pH 13); (**C**) Mott–Schottky plot of CuO@CuFe_2_O_4_ in 0.1 M NaOH (pH 13); (**D**) IPCE spectrum of CuO@CuFe_2_O_4_ measured at 0.5 V vs. RHE in 0.1 M NaOH (pH 13).

## Data Availability

The data presented in this study are available upon request from the corresponding author.

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
