# Peer review of "New Concept for the Facile Fabrication of Core–Shell CuO@CuFe2O4 Photocathodes for PEC Application"

_materials, 2022, doi:10.3390/ma15031029_

Round 1
Reviewer 1 Report
I have gone through the manuscript; my comments are given here.
- Authors are requested to arrange properly the introduction section, where they can discuss recent literature about the works, motivation, and novelty steps by step.
- Does figure 4 is normalized
- I didn’t find any characterization evidence of the core-shell structures as mentioned in the title.
- There should be a separate conclusion section.
- I found that the major focus was given on the characterization of the sample, rather than the explanation of the application point of view. More explanation of the discussion section will help to improve the quality of the manuscript.
Author Response
Dear reviewer, thank you for your valuable comment. Here are our responses:
- Information and literature has been added to the introduction.
- Figure 4 has been changed to avoid ant misunderstanding.
- XPS measurement and EDS mapping have been added to support the demonstration of core-shell structure.
- The body of the text has been rearranged
- Our priority was to show the synthetic approach which is novel and facile at the same time. Another important issue is characterization of the material itself in order to correlate it with the enhanced and kinetically improved PEC performance. Therefore, this is the core of the present article. However, a profound performance and mechanism is planned to be an objective of the forthcoming articles.
Reviewer 2 Report
In this paper, the authors report a thermal conversion preparation of CUFE Prussian blue analogues CuO@CuFe2O4 A new method of core-shell structure. The outstanding photoelectrochemical properties of the photocathodes of CuO@CuFe2O4 have been studied with use of combinatory photo-electrochemical instrumental techniques which proved the electrodes were stable over the whole water photolysis run under relatively positive potentials. There are some issues which the authors should address them before acceptance process of the paper. Here are my comments:
- The scales in Figure 1 are not clear, and the author needs to improve them.
- The author needs to prove that the nanostructure is a core-shell structure.
- In order to reflect the practicability of the film, the author needs to provide a photo of the sample.
- What are the advantages of this job over other jobs? The author is advised to make a table for comparison.
- As for high efficiency photocatalysts, some related literature authors need to mention, such as: https://doi.org/10.1007/s10562-020-03118-x; https://doi.org/10.1039/C7RA02198D; https://doi.org/10.1016/j.colsurfa.2021.127918
Author Response
Dear reviewer,
Thank you for your valuable comments. Here are our responses:
- Figure 1 has been improved as suggested.
- XPS measurement and EDS mapping have been added to support the demonstration of core-shell structure.
- Photos of the samples have been added into supporting information.
- The main advantage of our materials is its new synthetic approach with allowed to reach high level crystallinity and flexibility, and facile preparation. Moreover, in comparison to other reported systems, our system exhibited low recombination and high stability during irradiation.
- Suggested literature have been incorporated into the introduction.
Reviewer 3 Report
In this study, the authors prepared and investigated the PEC activity of CuO@CuFe2O4 films. In general, the article can be considered for publication after a major revision.
1) The manuscript should be polished by a native speaker. The purity of used reagents should be indicated for reproducibility purposes.
2) What was the thickness of obtained film? The film does not look uniform - supply large-scale cross-sectional SEM images. It will be also good to estimate average roughness by SEM or AFM.
3) XPS analysis should be done to investigate the oxidation states of Cu and Fe in prepared film.
4) To confirm the core-shell structure that authors should scratch a small amount of the film and investigate it under TEM. Otherwise, the formation of a "core-shell" structure is not obvious and this term should be avoided.
5) Figure 6A. Provide a dark-current scan of the prepared film.
6) It is not clear how the authors determined their parameters, why 0.3 V vs. RHE was used for the determination of max photocurrent density, and why 0.5 V vs. RHE was used for stability measurements!
7) The authors claimed that no photo-corrosion was observed but in reality decrease of photocurrent over time is clearly demonstrating the opposite view!
8) Introduction part can be improved by discussing some highly relevant studies such as DOI: 10.1016/j.jpcs.2016.07.014 and DOI: 10.1016/j.jallcom.2021.158724.
Author Response
Dear reviewer,
Thank you for your valuable comments. Here are our responses:
- English language has been thoroughly checked. The purity of used reagents has been added to the manuscript.
- Recently recorded cross-section images have been added to revised text.
- XPS measurement has been added.
- XPS measurement and EDS mapping have been updated to support the demonstration of core-shell structure.
- Figure 6 has been changed as suggested.
- We wanted to use a potential above onset potential to ensure the stability of the materials over time.
- The claim has been changed from no photocorrsion to small photocorrosion.
- Suggested literature has been incorporated into the introduction
Round 2
Reviewer 1 Report
Authors have improved the manuscript, it should be accepted.
Reviewer 2 Report
The article has been modified by the system and can be received.
Reviewer 3 Report
No more comments